# “I’m Not Who I Used to Be!” The Compelled Metamorphosing Process of Coping with Brucellosis Among Chinese Patients: A Qualitative Study

**DOI:** 10.3390/healthcare13010034

**Published:** 2024-12-28

**Authors:** Mei Zhou, Bo Zhu, Xueling Xiao, Xin Suo, Bo Fan, Honghong Wang

**Affiliations:** 1Xiangya School of Nursing, Central South University, Changsha 410083, China; 20120160@immu.edu.cn (M.Z.); zhubo2020@csu.edu.cn (B.Z.); xuelingxiao@csu.edu.cn (X.X.); 2School of Nursing, Inner Mongolia Medical University, Hohhot 010110, China; 20140031@immu.edu.cn (X.S.); 20220840@immu.edu.cn (B.F.)

**Keywords:** brucellosis, patient experiences, grounded theory coping, self-management

## Abstract

Background: Brucellosis, one of the most common zoonotic diseases globally, is a serious public health problem. The complex and diverse clinical manifestations pose numerous challenges for patients when coping with brucellosis. Scarce studies have been performed in China. Objectives: This study aimed to explore the process of coping with brucellosis and different aspects of the phenomenon from the perspective of patients, and propose a conceptual framework of patients’ processes in coping with brucellosis. Methods: Qualitative study based on constructivist grounded theory methodology using in-depth interviews and focus group discussions. The data analysis included initial coding, focused coding, and theoretical coding using the constant comparative method and memo writing. Results: The conceptual framework of “compelled metamorphosing” was constructed, which conceptualized three primary types of coping strategies: “blind persistence”, “resignation to avoidance”, and “proactive pacing”. Symptoms, financial strain, information cocoon, emotional value, and family responsibility had an important impact on coping strategies and played a significant role in driving their development. Conclusions: This paper provides new insight into patients’ lives and describes the strategies patients use to cope with the challenges and problems caused by brucellosis. Compelled metamorphosing represents a basic psychosocial process. These findings can be used to develop future complex interventions and studies.

## 1. Introduction

Brucellosis is an infectious bacterial disease caused by species of the genus *Brucella* spp. [1]. Due to the specific nature of its transmission pathways, the global pooled prevalence of brucellosis was 14% from 2000 to 2021 among livestock-related occupational groups [1]. In recent years, due to the dynamic growth of animal husbandry in China, opportunities for human infection have increased, leading to a continuous rise in the incidence of brucellosis, which has expanded to all provinces [2]. A total of 69,767 cases were reported by 2083 counties in mainland China in 2021, a 47.7% increase from 2020 (47,425) [3].

Clinical manifestations of brucellosis range from asymptomatic infections to acute, subacute, and chronic forms [4]. If it is not properly diagnosed in its acute phase and is left untreated, it can become chronic and persist for years. The chronicity of infection results in localization of the bacteria in various tissues and organs, causing debilitating complications, such as osteoarticular, hepatobiliary, central nervous system, and cardiovascular involvement [5,6,7,8]. However, due to the complex and diverse clinical manifestations of the illness, misdiagnosis, and delayed diagnosis often occur, causing serious adverse effects on patient health [9,10]. Even if patients receive timely treatment, they still need long-term treatment with multiple antibiotics and endure the side effects caused by antibiotics. Despite satisfactory antibiotic treatment, relapses and therapeutical failures are inevitable to a certain degree [9]. Thus, brucellosis is a disease that has a significant impact on quality of life [11], causes disability, and leaves permanent sequelae, depression, anxiety, loss of work hours, and medical expenses [2,12,13].

Patients with brucellosis are confronted with various challenges and problems. Coping as a dynamic process is defined as the cognitive and behavior strategies used to manage the external and/or internal demands of a specific person–environment transaction that is appraised as stressful, taxing, or exceeding the resources of an individual [14,15,16]. Studies have revealed that the way in which an individual copes with illness-specific and stressful life events may generally mediate the impact of disease severity on quality of life [17,18]. Therefore, understanding the process of how patients cope with brucellosis is necessary for health professionals to appropriately care for patients. However, earlier research has focused on quantitative studies of symptom types, frequencies, disease burdens, and quality of life [8,11,19]. However, how patients cope with the challenges of brucellosis and how different coping behaviors relate to personal well-being are underexplored. Therefore, the current study aimed to explore the process of coping with brucellosis and different aspects of the phenomenon from the perspective of patients, and propose a conceptual framework of patients’ processes in coping with brucellosis. This conceptual framework can benefit other stakeholders who are planning to provide more suitable interventions and programs for this population.

## 2. Materials and Methods

### 2.1. Design

To explain the process of coping with brucellosis and to provide a comprehensive explanation of the phenomena, constructivist grounded theory (CGT) was used in the study. CGT is a useful method for exploring and theorizing individual and social life [20,21]. CGT attends to researchers’ and research participants’ language, meanings, and actions. It can start from the inside to understand research participants’ meanings and actions [21]. Using this method affords a detailed investigation of the behaviors and thoughts of individuals faced with brucellosis and learning about their coping processes.

### 2.2. Setting

The studies were conducted at the International Hospital of Mongolian Medicine (IHMM), which is located in Hohhot, the capital city of the Inner Mongolia Autonomous Region (IMAR) of China. The IMAR is one of the most critical areas for livestock husbandry, a historically endemic area of human brucellosis in China [22]. The IHMM is a comprehensive hospital that has been designated as a specialized institution for the treatment of brucellosis. Mongolian medicine treatment of brucellosis has a long history, and it is one of the important methods for the prevention and control of brucellosis in IMAR [23]. Currently, the Department of Brucellosis at IHMM has achieved significant therapeutic effects by utilizing Mongolian medicine and antibiotics in the treatment of brucellosis [23]. The patients treated in the Brucellosis Department of the IHMM come from various parts of the country, indicating geographical and sociocultural diversity in their experiences of the illness.

### 2.3. Sampling and Recruitment

To recruit patients with brucellosis, we contacted the head of the Brucellosis Department and requested their assistance in inviting eligible patients. The inclusion criteria were as follows: (1) all patients who met the relevant diagnostic criteria for brucellosis in the Brucella Disease Diagnostic and Treatment Guidelines, with a diagnosis of brucellosis confirmed by pathological examination [24]; (2) patients aged 18 years or older; and (3) patients capable of understanding and speaking Mandarin Chinese. Individuals who had a mental illness or were unable to communicate effectively were excluded.

During the initial stage of this study, purposive sampling was used to recruit participants who could provide rich information. After 27 interviews, categories were tentatively established. Theoretical sampling was used to collect pertinent data to elaborate on the categories and their properties [20]. Sampling was discontinued upon reaching theoretical saturation, which was determined based on specific criteria. These included the absence of new themes or codes in the final three interviews, redundancy in findings as previously identified themes were consistently reiterated, and a point where variations within themes no longer significantly expanded the understanding of the concept.

### 2.4. Data Collection

Two types of data collection methods were used: focus group discussions and in-depth interviews. From November 2021 to October 2022, 37 face-to-face in-depth interviews and 1 focus group discussion (with 4 patients) were conducted by MZ or XS or BF. Prior to conducting the interviews, all interviewers underwent training sessions to familiarize themselves with the study objectives, the interview guide, and effective qualitative interviewing techniques. This training included role-playing exercises and mock interviews to align the researchers’ approaches. The research team held regular meetings throughout the data collection process to discuss ongoing interviews, address any deviations or challenges, and ensure alignment in interpreting and implementing the interview guide. Initially, the interview questions were open and broad, such as “Tell me about what happened to you after you contracted brucellosis?” Participants were asked to talk as widely as possible about their process of coping with brucellosis. With the development of the emerging categories, the interview questions became more focused. Table 1 provides examples of the questions asked during the interviews.

The focus group consisted of four patients from different regions who were relatively familiar with each other. The discussion was guided by a moderator, aiming to stimulate conversation and explore different viewpoints. The moderator posed open-ended questions to encourage participants to engage in in-depth discussions while recording the group’s interactions. The purpose of the focus group was to gather shared experiences and diverse perspectives.

The interviews were digitally recorded and transcribed verbatim for continuous data analysis and interview development. Participants’ anonymity was maintained using a coding system.

### 2.5. Data Analysis

All data were transcribed verbatim, and the participants were given pseudonyms for anonymity. The data analysis included initial coding, focused coding, and theoretical coding using the constant comparative method (compare incident with incident, and memo writing).

Initial coding involved naming each word, line, or segment of data. The codes were closely related to the data, showed actions, and indicated the progression of events from the participant’s point of view. During the initial coding, MZ and BZ (female, PhD candidate, extensive knowledge of qualitative research) conducted line-by-line coding. The goal was to remain open to all possible theoretical directions indicated by the researcher’s readings of the data [20]. After we established some strong analytic directions through the initial coding, we began focused coding to synthesize, analyze, and conceptualize larger segments of the data. Focused coding requires decisions about which initial codes make the most analytic sense to categorize the data incisively and completely [20]. Subsequently, theoretical coding was used to theorize the data and specify possible relationships between the categories [20].

The qualitative data were managed using NVivo12 (12.2.0.443, QSR International Pty.Ltd, Doncaster, Victoria, Australia). Each transcript was stored within NVivo, which supported the data analysis. NVivo was used following line-by-line coding in Word documents in an attempt to manage the vast array of collected data. NVivo provided an audit trail, adding rigor to the analysis process. Memo writing was utilized throughout the research process. Memos were written freely and immediately as ideas occurred. Memo writing creates an interactive space for conversing with researchers about their data, codes, ideas, and hunches, and provides a record of their research and of their analytic progress [20]. A sample memo can be found in Box 1. Discrepancies were resolved through discussion and consensus. The core research team (MZ, BZ, XS, and HW) met regularly to review the ongoing analysis.

Box 1.Memo—‘Emotional value’  Patient 8 is experiencing severe fatigue symptoms. She is unable to handle most of the household chores at home. Consequently, their mother-in-law came to help take care of her two children. Unfortunately, she does not feel relaxed with the arrangement; instead, she feels very frustrated and unhappy. Her mother-in-law and husband complain and frequently get angry with her. This illustrates that “family support” is important for the patient. (10 January 2022)  Patient 18’s wife said that the patient is easily irritated, and frequently expresses anger, which in turn causes her to become angry as well, especially during the hospitalization. As a result, she does not want to accompany him to the hospital. This, in turn, affects the patient’s mental state and treatment progress. From this incident, it can be observed that the emotions between the patient and the relatives are mutually influential.The term “family support” often emphasizes one-sided support from family members to the patient. However, through multiple interviews, we found that the patients and their relatives, when faced with this illness, failed to provide each other with positive emotional value. Therefore, using the term “family support” to encode this event may be too broad and lack precision. The code “emotional value” may be more accurate in describing the situation. (29 March 2022)

### 2.6. Rigour and Trustworthiness

To ensure a deep understanding of the participants’ experiences, the researchers spent sufficient time with the patients, fostering trust and rapport. We engaged in regular meetings within the research team to discuss the data collection and analysis process. This collaborative approach helped ensure consistency in data interpretation and addressed any potential biases or inconsistencies in the research process. The researchers documented their intuitions, suspicions, feelings, and thoughts throughout the research process by writing memos. This approach aimed to minimize potential biases that could arise during interviews and analysis, ensuring that the conceptualization of the phenomenon is closely tied to the data.

### 2.7. Ethical Considerations

Ethical approval for this study was obtained from the Xiangya School of Nursing, Central South University (approval code: E202171, approval date: 24 June 2021). All participants were adults (aged 18 years and above) and provided signed informed consent prior to their participation. They were also informed of their right to withdraw from the study at any time without any repercussions. The informed consent process was conducted in a private setting to eliminate any sense of coercion. The interviews were carried out by trained researchers who fostered a respectful and empathetic environment to minimize potential psychological discomfort. To ensure confidentiality, all interview transcripts and recordings were anonymized, and all identifying information was securely stored and accessible only to authorized personnel.

## 3. Results

### 3.1. Participants’ Characteristics

Forty-one participants (29 males and 12 females) aged 19–68 years were included in the study. Patient interviews took place either in the health education room (*n* = 32) or in the patient’s ward (*n* = 9). All interviews were conducted in Chinese. They were audio-recorded and lasted from 17.50 to 59.23 min, with a median of 32.15 min. The participants’ socio-demographic and clinical characteristics are summarized in Table 2.

The overall process of coping with illness among the participants is shown in Table 3. The conceptual framework of “compelled metamorphosing” was constructed, which conceptualized three primary types of coping strategies: “blind persistence”, “resignation to avoidance”, and “proactive pacing”. These three types of relationships are compatible and not independent; they can coexist or transition with each other in certain contexts (Figure 1). “Blind persistence” and “resignation to avoidance” emerge with adverse effects on patients’ health outcomes, whereas “proactive pacing” is associated with positive health outcomes for patients. In addition to the symptoms influencing patients’ coping behaviors, this study highlights the importance of financial strain, information cocoon, emotional value, and family responsibility in the process of coping with illness among patients.

### 3.2. Compelled Metamorphosing

“I’m not who I used to be!” was the most commonly spoken phrase by patients. The core category in this study was “compelled metamorphosing”. Compelled metamorphosing was defined as significant and unavoidable behavioral changes that occur in patients while coping with brucellosis due to the impact of the illness. It emphasizes the transformative nature of their experience, which is not of their own choice but is compelled by the challenges posed by the illness. Compelled metamorphosing has many types, with different classifications depending on patients’ behaviors in adhering to medical treatment, engaging in activities, and participating in social interactions during the process of coping with brucellosis.

#### 3.2.1. Blind Persistence

Blind persistence refers to the behavior of patients who disregard medical advice or the impact of illness on the body, instead persisting in their preexisting lifestyle. This phenomenon often occurs in patients with relatively mild symptoms. Specific manifestations of patients’ behaviors included non-compliance with medical treatment, activity persistence, and maintenance of the existing social circle.

Non-compliance with medical advice. This refers to patients intentionally deciding not to comply with the prescribed medical treatment plan or regimen. This behavior stemmed from various reasons, such as patients’ information cocoons leading to misperceptions about brucellosis.

Patient 11, a 22-year-old male, refused conventional treatment following illness onset because he firmly believed that long-term medication would be harmful to his health and considered brucellosis an incurable condition despite repeated health education provided by medical staff. He only sought medical attention when his condition was severe.


*Patient 11: If you take so many medications, what if this disease is cured, but another one pops up? … There’s a guy in our village who got treated at the hospital and then continued taking medication at home for years; he ended up with uremia.*



*Interviewer: Do you know him? Is his uremia indeed a result of taking these medications?*



*Patient 11: I don’t know him personally; I just heard about it … Honestly, I’ve never seen anyone get cured … Even if it’s treated successfully, it’s likely to come back. Since it is incurable, there is no need for treatment.*


Non-compliance with medical advice can be harmful, as it may lead to worsening of the condition, complications, or missed opportunities for early intervention and effective management.

Activity persistence. It entailed the patients’ unyielding pursuit of activities, as they would have done before falling ill despite being unwell, disregarding their body’s tolerance and potential harm caused by these activities.

*I usually rest only after completing all my tasks. I can endure a lot. Even if my body is in pain, I still have to finish my work; otherwise, I can’t sleep.* [P18, male]

However, persisting in activities sometimes exacerbates the condition, which can result in the patient entering an overactivity–underactivity cycle, which is a pattern in which periods of excessive activity alternate with prolonged rests [25]. Such a cycle was observed in Patient 4, a 59-year-old farmer who had been suffering from aortic valve insufficiency for several decades. She was usually asymptomatic, but her fatigue symptoms worsened after contracting brucellosis. Within the previous six months prior to the interview, she was transported to the hospital three times by ambulance, with two of those instances occurring after her discharge, upon which she had failed to follow medical advice and had overexerted herself, leading to a worsening of her condition and readmission.

Maintenance of existing social circles. In China, interpersonal relationships among friends are often maintained through communal meals and social drinking activities. To maintain their existing social circles, some patients compromised their health by concealing their medical conditions and continuing to engage in activities such as drinking alcohol or consuming spicy foods, despite medical advice advocating abstinence from alcohol and avoidance of spicy foods.

#### 3.2.2. Resignation to Avoidance

Resignation to avoidance implies that patients with brucellosis submitted to the behavior of avoiding certain aspects of their medical condition or treatment without actively seeking solutions or support to change it. This was often due to feelings of helplessness. Patients in a state of resignation to avoidance exhibited the following three behaviors: symptom-driven treatment behavior, activity avoidance, and withdrawal from social circles.

Symptom-driven treatment behavior. Symptom-driven treatment behavior was one of the common ways in which patients with brucellosis coped with symptoms. Some patients, despite knowing that adhering to proper treatment for brucellosis can lead to a cure, only sought medical treatment when symptoms became severe enough to impact their daily lives, often due to sociocultural factors, such as economic conditions or family responsibilities.

*I am exhausted after a day’s work, to the point where I can’t endure it anymore. I feel extremely fatigued… I need to take a day off after a day of work. It has reached a point where I can’t bear it anymore, and I am experiencing severe swelling. That’s why I came (to the hospital) again. Otherwise, I wouldn’t have come.* [P5, male]

Activity avoidance. Activity avoidance refers to the patient’s act of consciously refraining or staying away from engaging in certain activities or tasks. “I can’t do anything” was the most common phrase used by patients with brucellosis, and it could have three different meanings in different patients. Patients in the acute phase of the illness, experiencing severe pain and fatigue, often felt “I am incapable of doing anything.” However, patients in the chronic phase of the disease often experienced feelings of “I don’t want to do anything” and “I’m afraid to do anything.” “I don’t want to do anything” was related to the fatigue symptoms experienced by the patients. Brucellosis, known as the “lazy man’s disease” in Inner Mongolia, was characterized by patients expressing a lack of motivation to engage in any activities due to fatigue. “I’m afraid to do anything” was commonly heard in patients with chronic pain. After experiencing severe pain, they were afraid of engaging in activities that may cause pain again, which is also known as kinesiophobia [26,27].

*I can’t do anything anymore. I can’t do it … If I do (anything), I might lying there in pain, and then (my family) will have to take me to the hospital. It’s just more trouble.”* [P37, male]

Activity avoidance during the acute phase of the illness can help patients prevent worsening symptoms. However, in the chronic phase, prolonged activity avoidance can lead to feelings of uselessness and helplessness in patients.

Withdrawal from social circles. Withdrawal from social circles is defined as a patient’s intentional reduction, or even refusal, of social participation, gradually resulting in the individual’s withdrawal from their social circles. In this study, symptom-induced physical discomfort was identified as a significant factor contributing to patients’ withdrawal from social circles.

*If I’m having dinner with friends, I simply can’t join in without drinking. I just sit there, and it becomes a bit uncomfortable. Oh, I’m the buzzkill… In addition, I feel tired all the time, wherever I go, I just want to sleep. Can I go to a friend’s house and just lie down the whole time? No, you can’t! You can only lie down at home.* [P41, female]

For some patients, withdrawal from social circles was an involuntary decision, as brucellosis is an infectious disease. Discrimination and rejection from others prevented patients from integrating into social circles, leaving them resigned to withdrawal from social circles.

*In my community, all the neighbors know that I have contracted brucellosis. As soon as I appear, they disperse (afraid of getting infected by me). It’s true. They scatter as soon as I show up.* [P36, male]

#### 3.2.3. Proactive Pacing

Proactive pacing refers to the patient pacing themselves to find the right speed or rhythm for their daily life so that they can preserve energy to do more important tasks. Specific behaviors of pacing encompassed pacing to comply with medical advice, activity pacing, and rebuilding of new social circles.

Pacing to comply with medical advice. This describes patients’ adherence to the prescribed treatment plan by adjusting their lifestyle rhythm and pace, which includes following medication regimens and attending medical appointments as instructed.

*I’m here for the sixth time (hospitalization). I live quite far from here (Inner Mongolia), over a thousand kilometers away, so it’s not easy to come here, and every time I get discharged, I make sure to take home a three-month supply of oral medications … About every 21 days or so, I go to the local hospital to get my liver and kidney function checked. Those powdered Mongolian medicines taste really bitter, after taking the medicine, I rinse my mouth and have a candy, that’s how I make it through … When I’m close to finishing it, I’ll contact the doctor, briefly describe my situation. He’ll arrange a bed for me, and I arrange work and home matters, then I’ll come here.* [P23, male]

Activity pacing. Some patients adopted specific strategies based on their physical condition to adapt their activities, enabling them to accomplish certain tasks without imposing further strain on their bodies. The most frequently used methods for pacing included changing position to time, avoiding heavy physical activity, slowing down, and breaking up activities into manageable pieces.

*When doing (work), I consider dividing the tasks that were originally planned for a day into three, four, or five days (to complete), and work appropriately. It’s not like before when I could finish the workload of two days in just one day.* [P41, female]

Rebuilding of new social circles. For patients, this implies the reconstruction of new social connections after experiencing isolation or withdrawal from their existing social circles. It involved actively forming new friendships, and joining social groups to establish a sense of belonging and support within their social environment. For instance, some patients admitted during the same time period established WeChat groups, wherein they engaged in reciprocal communication, encouragement, and assistance.

*We’ve created a WeChat group called “Embarking Toward Happiness and Health”. In the group, we help each other out, ask if there are available beds in the ward, and see if someone is recovering faster. When we witness others getting better, it gives us hope that we can also recover soon.* [P7, male]

### 3.3. Sociostructural Factors

In this study, the intricate behaviors of the patients were influenced not only by the direct effects of the symptoms but also by the following sociocultural factors: financial strain, information cocoons, emotional values, and family responsibilities.

Financial strain. Financial strain was one of the significant sociocultural factors affecting patients’ medical adherence behavior. In this study, patients with better medical adherence were often those facing less financial strain, such as ranch employees. This was because contracting brucellosis due to occupational hazards qualified as a work-related injury, and the ranch covered the treatment costs.

*This is my sixth hospitalization. Each hospitalization costs over 10,000 yuan, and if medication is prescribed upon discharge, it’s nearly 20,000 yuan. When I return to work, the company reimburses all expenses, including transportation costs. However, for others like them (other patients) who are farmers, spending over 10,000 yuan for hospitalization is a huge economic burden.* [P23, male]

However, for farmers and herders, although the government partially reimbursed medical expenses, brucellosis not only reduced the patients’ earning capacity but also caused economic losses when their livestock contracted brucellosis. Consequently, they faced greater financial strain, leading to lower medical adherence compared to ranch employees.

*Outpatient expenses aren’t covered by medical insurance, but a portion of the expenses incurred during hospitalization can be reimbursed. That’s why I only get medication when admitted. After discharge, if the medication runs out, I just let it be. It costs money to be hospitalized, and I don’t earn much. I’m just managing to maintain our household…I couldn’t walk anymore, so I had to go to the hospital; otherwise, I wouldn’t be here.* [P5, male]

Information cocoons. This study found that the patients had various avenues to acquire knowledge about brucellosis, commonly including online searches, consulting doctors, and seeking advice from fellow patients. During this process, patients were prone to forming information cocoons. The information cocoon is described as a situation in which patients exclusively obtain information on the topic they care about and from the perspective they identify with, avoiding information that holds opinions different from theirs. Over time, they become ensnared in the information cocoon, even though their opinions about certain events are unconsciously confined. This is detrimental to the patients’ awareness of brucellosis and influences coping behavior toward the illness. For instance, as mentioned earlier, Patient 11 heard about a villager who developed uremia due to long-term medication. This coincides with his preconceived notion that ‘all medicines contain three-tenths of poison.’ Consequently, without verifying the authenticity of this story, he stubbornly believed that long-term medication would be detrimental to his health, despite being informed by doctors that brucellosis can be cured through proper treatment. As a result, he refrained from complying with medical advice and only came to the hospital when his symptoms worsened significantly.

The family’s information cocoons can also have an impact on the patient. For instance, Interviewee 9’s wife, even after being informed about the rarity of person-to-person transmission of brucellosis, remained resolute in her belief that since it is an infectious disease, isolation measures should be taken. Consequently, this led to the patient developing negative emotions.

*I get easily annoyed now. Yesterday, I even had a fight with my wife. She isolated me; she separated the dishes and plates at home. Even my child’s homework book, she says, “Don’t touch it!” …Even though the doctor said it wasn’t necessary, she insists that since brucellosis is an infectious disease, it’s necessary. Especially in the current COVID-19 situation, everyone is more sensitive to infectious diseases.* [P9, male]

Emotional values. Emotional value is the difference between an individual’s perceived positive emotional experiences and negative emotional experiences when interacting with others; therefore, emotional value can be either positive or negative. This study revealed that the majority of patients’ family members often excelled in providing tangible support (concrete assistance, such as undertaking household chores) in daily life, while frequently overlooking offering positive emotional value to the patients.

For example, Patient 8 was resting at home due to severe fatigue. Although her family assisted with the majority of household chores, they provided negative emotional value. Consequently, she remained unhappy, felt unjustly treated, and experienced depression.

*After I got diagnosed, his reaction and attitude were explosive, like a pot about to boil over… He has a really short temper … He said, “Out of all the people at the factory, why did this have to happen to you?” That’s how he was. He wouldn’t talk to me properly. He’d just keep complaining and blaming me for everything … I just don’t understand why everything seems annoying, and nothing sounds right.* [P8, female]

The provision of emotional value was bidirectional; if the patient was unable to offer positive emotional value to the family members, it could directly influence the patient’s treatment process.

Family responsibility. Family responsibility was an important factor that influenced the patients’ behavioral choices. Some patients feared that falling ill and being hospitalized would burden their family, so they chose to silently endure the discomfort brought on by the illness until the symptoms became severe and unbearable, at which point they decided to seek medical attention.

*I didn’t even tell her (my daughter) when I was feeling sick; I’m afraid it would disrupt her work… It got to the point where I couldn’t even walk, and my daughter had to carry me to the hospital … So, that’s why I say it caused a delay in treating (the illness).* [P34, female]

By contrast, some patients perceived themselves as the backbone of the family and admonished themselves not to be defeated by illness. Therefore, they strove to actively confront the disease.

*My current mindset is just to get better as quickly as possible. Once I’m cured, then we can talk about other things. I’ve hardly forgotten about things like taking medication on time. I prioritize this illness. Because right now, we’re in a phase where we have elderly parents and young children to take care of, you know? At the very least, I need to get better. Even if I can’t take care of the elderly, I have to take care of the kids ourselves, at the very least.* [P8, female]

### 3.4. The Relationship Between Categories and Subcategories

The patients’ coping with brucellosis was a complex process, with symptoms directly influencing their behavior. Symptoms that had a greater impact on patients tended to result in better medical adherence, while those with a lesser impact seemed to lead to poorer adherence to medical advice. Additionally, sociocultural factors significantly influenced the patients’ coping behaviors. Patients who experienced less financial strain, obtained positive emotional value in social interactions, and considered family responsibility to be a driving force were more likely to transform into “proactive pacing”. By contrast, patients who faced greater financial strain, were trapped in information cocoons, often experienced negative emotional values in social interactions, and perceived family responsibilities as burdens were more prone to exhibiting changes toward “blind persistence” and “resignation to avoidance”. Therefore, under the influence of various factors, the three types of compelled metamorphosing states were not only transformative but also exhibited cross-cutting behaviors (Figure 1). Overall, “proactive pacing” represented a positive coping process that contributed positively to the patients’ health outcomes, while “blind persistence” and “resignation to avoidance” represented negative coping processes that were detrimental to the patients’ health outcomes.

## 4. Discussion

To our knowledge, this is one of the first studies to use a CGT approach to explore the process of coping with brucellosis and to construct an interpretive understanding from the patients’ perspective. Previous literature and research have not addressed the process of coping with brucellosis, focusing instead on the direct impact on patients’ quality of life and psychological well-being, as well as the effects of different treatment approaches on the disease outcome [9,11,13]. The conceptual framework presented herein situates compelled metamorphosing as central to patients’ coping processes with brucellosis. Blind persistence, resignation to avoidance, and proactive pacing represent the three primary response types adopted by patients in coping with the illness. In addition to symptoms directly affecting patients’ coping processes, financial strain, information cocoons, emotional values, and family responsibilities emerge as primary sociocultural factors.

Compelled metamorphosing was inevitable for brucellosis patients and was characterized by changes that were forced upon them. Our findings are in line with Bury’s contention that illness, especially chronic illness, is a disruptive event that leads to changes in the structure of daily life [28]. “Compelled metamorphosing” is also similar to the idea of the Chronic Illness Identity Model, which posits that the diagnosis and ongoing experience of chronic illness can lead to the reshaping of an individual’s self-concept and identity [29]. In the face of brucellosis, patients with ‘blind persistence’ coping style became adversaries to the illness, denying or evading it, manifesting as refraining from seeking medical treatment, activity persistence, and sustenance of social circles. This coping behavior can also be referred to as ‘illness denial’, a similar coping strategy reported in other infectious diseases, such as acquired immune deficiency syndrome (AIDS) and COVID-19 [30,31,32,33,34,35]. Similar to the findings of this study, previous research indicates an association between this coping strategy and adverse clinical outcomes [30,31,35,36]. Patients with a ‘resignation to avoidance’ coping style became slaves to the illness, losing control over their lives. This passive state is not only detrimental to recovery from the disease, but can also lead to psychological issues. For example, prolonged ‘activity avoidance’ may cause patients to perceive themselves as worthless individuals who burden their families, resulting in negative emotions. As also observed in other diseases characterized primarily by pain, activity avoidance was associated with negative psychosocial outcomes [37,38,39]. By contrast, patients with the ‘proactive pacing’ coping style treated the illness as a new friend, learning to coexist harmoniously with it. A study investigating the experiences of patients with inflammatory bowel disease, which involves a long-term treatment process similar to brucellosis, reported similar results, showing that patients use “reconstruction of individual, social, and professional life” as their main strategy for better illness management [40]. The patients experienced many problems and worries after being affected by brucellosis. The coping strategy of pacing helped the patients seek maximum adaptation to the changes brought about by the illness, minimizing their problems and managing the illness. Compared to denying and passively dealing with illness, pacing can result in a less negative impact of illness and a greater sense of control over the priorities of daily life, which is consistent with previous studies [40,41].

In addition to the symptoms influencing patients’ coping strategies, this study also identified some sociocultural factors, namely financial strain, information cocoon, emotional values, and family responsibilities. The “Information Cocoon” was first proposed by the American scholar Sunstein in the book “Information Utopia” in 2008. It is defined as a special “communication universe” where people only choose to hear information that makes them comfortable and pleasant [42]. Nowadays, there are multiple avenues for patients to acquire knowledge about brucellosis. For some patients, when the information they acquired overlaps highly with their previous ones, they tend to focus on homogeneous information and may even resist other different but accurate or more meaningful information, thereby creating an “information cocoon”. Research has shown that information cocoons can deepen peoples’ inherent biases, leading to their self-perception bias and irrational inflation, making them prone to form radical and extreme views, statements, or behaviors [43]. A systematic review showed that awareness and knowledge of brucellosis are insufficient, and a lack of knowledge has been reported to result in inappropriate health behavior in response to brucellosis-like symptoms [44,45,46,47]. Our findings are consistent with these previous studies and provide unique insights into the causes of inadequate or even erroneous cognition of brucellosis, suggesting that healthcare professionals assess whether patients are trapped in information cocoons before providing health education and take measures to break the information cocoons if present.

Previous studies have found that emotional value is associated with the person’s behavior, quality of life, and psychological aspects of well-being [48,49]. Similarly, we found that positive emotional values can bring patients good feelings, stimulate positive emotions, and encourage them to actively cope with the illness; conversely, negative emotional values bring negative emotions to patients, leading to their passive coping with the disease. Furthermore, because of the emotionally dyadic nature of emotional value [48], focusing on the emotional value of both the patients and the interactants (such as caregivers and medical professionals), rather than just one or the other, is a better way to maximize emotional value.

According to our participant experience, family responsibilities function as a double-edged sword. Some of the patients exhibited delayed healthcare-seeking behavior due to these obligations, resulting in feelings of guilt for becoming a drain on the family, whereas others perceived the necessity of actively treating their illness and minimizing its impact on themselves and their families, driven by their sense of family duty. Similar findings regarding family responsibilities have been reported in other studies. Ahmed et al. found that family responsibilities were facilitators of adherence to antiretroviral therapy among people living with HIV/AIDS [50], and a study of female patients with cardiovascular disease showed a high burden of family responsibilities as a barrier to cardiac rehabilitation [51]. The dual impact of family responsibilities on patients’ coping strategies may be attributed to Confucian values, which have deeply influenced Chinese society. These values prescribe distinct gender roles, traditionally positioning men as primary breadwinners, while women are expected to assume domestic responsibilities [52]. In particular, when men are diagnosed with brucellosis, they often recognize the importance of good health for maintaining their earning capacity, prompting them to adopt more proactive coping strategies. Conversely, women tend to prioritize caregiving for family members, often neglecting their own health and resorting to more passive coping strategies. Therefore, understanding the main influencing factors affecting the coping strategies of patients with brucellosis and promoting the adoption of positive coping strategies through interventions may be an effective and promising strategy for patients with brucellosis in the future.

This study reveals the process of coping with illness among patients with brucellosis through the development of a concept framework, as well as the factors influencing their coping behaviors. Specifically, the dual impact of emotional value and family responsibilities on patients’ coping behaviors has not been explored in previous studies on the coping behaviors of patients with other infectious diseases. The results provide theoretical support for tailored interventions to improve patient clinical outcomes.

There are several limitations to the present study. First, the data were collected at a Mongolian Traditional Medicine hospital, where participants received both antibiotic treatment and traditional Mongolian medicine, impairing the generalizability of the findings. Second, the study participants were asked to talk about an event that had happened in the past, which could have led to recall bias.

## 5. Conclusions

Coping with brucellosis is a complex process that changes as it progresses. We have identified three core types of coping strategies: “blind persistence”, “resignation to avoidance”, and “proactive pacing”. Each of these strategies influences patients’ health outcomes in different ways. Furthermore, the study highlights several external factors beyond symptoms, such as financial strain, information cocoon, emotional values, and family responsibilities, that play significant roles in shaping patients’ coping behaviors. These factors contribute to the complexity of the coping process and can either exacerbate or support the patients’ ability to manage the disease effectively. The findings provide a comprehensive understanding of the coping process of patients with brucellosis and inform future interventions targeted at the coping process to enhance better outcomes among this population.

## Figures and Tables

**Figure 1 healthcare-13-00034-f001:**
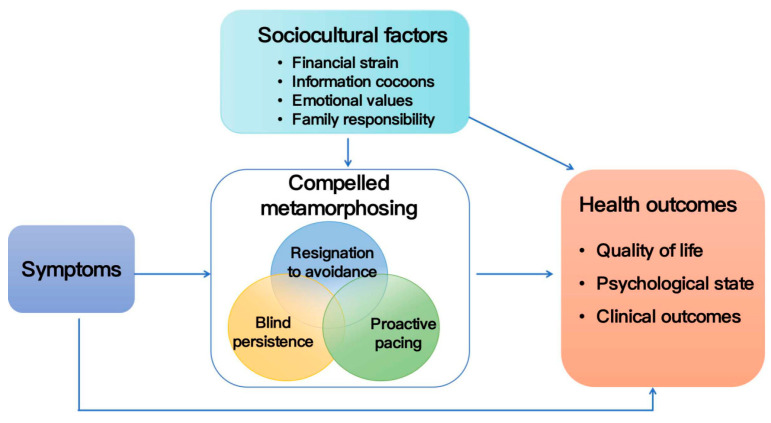
Conceptual framework of compelled metamorphosing.

**Table 1 healthcare-13-00034-t001:** Examples from the topic guide.

Categories	Questions
Initial open-ended questions	Tell me about what happened to you after you contracted brucellosis?
	Tell me about your experiences living with the disease.
	What impacts has the disease had on your life?
Intermediate questions	How do you cope with brucellosis?
	What helps you cope with the illness?
	What problems might you encounter? Could you tell me the sources of these problems?
Ending questions	After having these experiences, what advice would you give to someone who has just discovered that he or she contracted brucellosis?
	Is there anything you would like to ask me?

**Table 2 healthcare-13-00034-t002:** Socio-demographic characteristics of the participants (*n* = 41).

Characteristics	*n* (%)	Characteristics	*n* (%)
Gender		Age	
Female	12 (29.3)	20–29	5 (12.2)
Male	29 (70.7)	30–39	10 (24.4)
Ethnicity		40–49	6 (14.6)
Han	38 (92.7)	50–59	11 (26.8)
Mongolian	2 (4.9)	60–69	8 (19.6)
Manchu	1 (2.4)	70–79	1 (2.4)
Marital status		Occupation	
Married	39 (95.1)	Farmer	14 (34.1)
Widowed	1 (2.4)	Pastoralist	2 (4.9)
Unmarried	1 (2.4)	Milk handlers	3 (7.3)
Resident		Veterinarian	7 (17.1)
Living in IMAR *	15 (36.6)	Meat vendor/Abattoir worker	2 (4.9)
Living outside IMAR	18 (43.9)	Other ranch employees	6 (14.6)
Education background		Others	7 (17.1)
Primary school and below	10 (42.4)	Cognition of brucellosis	
Junior high school	17 (41.5)	Unknown	7 (17.1)
Senior high school	11 (26.8)	Heard of, with limited knowledge	22 (53.7)
University	3 (7.3)	Adequate knowledge	2 (4.9)
Duration of brucellosis infection(months)		Route of contracting brucellosis	
0–6	23 (56.1)	Unknown	7 (17.1)
7–12	5 (12.2)	Contact with cattle and sheep	31 (75.6)
over12	13 (31.7)	Contact with brucella vaccines	3 (7.3)

* IMAR = Inner Mongolia Autonomous Region.

**Table 3 healthcare-13-00034-t003:** Overall process of coping with illness among Chinese patients living with brucellosis.

Core Category	Categories	Subcategories
Compelled metamorphosing	Blind persistence	Non-compliance with medical advice
		Activity persistence
		Maintenance of existing social circles
	Resignation to avoidance	Symptom-driven treatment behavior
		Activity avoidance
		Withdrawal from social circles
	Proactive pacing	Pacing to comply with medical advice
		Activity pacing
		Rebuilding of new social circles

## Data Availability

Data used for this study are available from the corresponding author on reasonable request.

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
