# Peer review of "“I’m Not Who I Used to Be!” The Compelled Metamorphosing Process of Coping with Brucellosis Among Chinese Patients: A Qualitative Study"

_healthcare, 2024, doi:10.3390/healthcare13010034_

Round 1

Reviewer 1 Report

Comments and Suggestions for Authors

Dear Authors,

This study provides valuable insights into the framework for coping with the challenges faced by patients with brucellosis, utilizing the Grounded Theory method. I found it particularly interesting that the study also refers to the cultural influences on coping behaviors in China.

I commend the authors for their thorough analysis of detailed interviews conducted with more than 40 participants.

Below are some points of concern:

Ethical Considerations: I could not find any mention of ethical considerations within the manuscript. Given that the subjects are patients and thus vulnerable, mere mention in the informed consent section seems insufficient. Please include in the manuscript how voluntariness was ensured and protection was provided for the study participants.

Conclusions: While the current conclusions are satisfactory, using the identified (core) categories to explain the coping processes would likely make the content clearer and more comprehensible to the readers. Please consider this suggestion.

Thank you.

Best regards,

Author Response

Thank you for the insightful suggestions, please see the attachment.

Reviewer 2 Report

Comments and Suggestions for Authors

MANUSCRIPT EVALUATION REPORT

I. General Considerations

Strengths:

1.     The study addresses a pertinent and under-explored topic: the coping process of brucellosis among Chinese patients.

2.     The selected qualitative methodology (constructivist grounded theory) aligns appropriately with the study objectives.

3.     The manuscript exhibits a clear and well-organized structure.

4.     Results are presented in detail and substantiated by direct participant quotations.

5.     The study offers valuable insights into the experiences of brucellosis patients, which may inform clinical practices and health policies.

Weaknesses:

1.     Less than 50% of the references are from the past five years, indicating a potential lack of currency in relation to the most recent literature.

2.     There is limited discussion of studies specifically addressing coping with brucellosis.

3.     The Chinese cultural context could be more extensively explored in the discussion of results.

Suggestions for Improvement:

1.     Update references to include more recent studies on brucellosis and coping strategies in infectious diseases.

2.     Incorporate more studies specifically focused on coping with brucellosis.

3.     Expand the discussion on the Chinese cultural context and its impact on disease coping.

4.     Consider including recent systematic reviews or meta-analyses on brucellosis or coping with chronic diseases.

II. Section-by-Section Evaluation

1.     Title and Objective

Strengths:

·        The title clearly reflects the study content.

·        The objective is well-defined and consistent with the title and the investigated problem.

Suggestions for Improvement:

·        No significant suggestions for this section.

2.     Methodology

Strengths:

·        The methodological strategy (CGT) is appropriate for the study objectives.

·        The method is well-described, including information on participants, data collection, and analysis.

·        Ethical aspects were duly considered and reported.

Weaknesses:

·        Details about the theoretical saturation process are lacking.

·        There is limited information on how the focus group was conducted and integrated with individual interviews.

Suggestions for Improvement:

·        Provide more details on the theoretical saturation process.

·        Explain how the focus group was conducted and how its data were integrated with individual interviews.

·        Describe more thoroughly the process of developing the interview topic guide.

3.     Data Analysis

Strengths:

·        The analysis is well-articulated with the objectives and the CGT theoretical framework.

·        The analytical process is clearly described, including initial, focused, and theoretical coding.

·        The use of software (NVivo12) for data management is mentioned.

Suggestions for Improvement:

·        Provide more details on how analytical decisions were made during the coding process.

·        Explain more thoroughly how theoretical saturation was achieved.

4.     Results

Strengths:

·        The main results are highlighted clearly and objectively.

·        The use of tables and figures effectively complements the text.

·        Direct quotations from participants effectively illustrate the identified categories.

Suggestions for Improvement:

·        No significant suggestions for this section.

5.     Discussion

Strengths:

·        The discussion is pertinent and sufficient, contextualizing the findings within existing literature.

·        There is an in-depth analysis of the identified coping strategies.

·        The practical implications of the findings are well-discussed.

Weaknesses:

·        There is a lack of in-depth discussion on how the findings compare specifically with previous research on brucellosis.

Suggestions for Improvement:

·        Include a more detailed discussion on how this study's findings compare specifically with previous research on brucellosis.

·        Reflect more on how the results might be applied in different cultural contexts.

6.     Conclusions

Strengths:

·        The conclusions are clear and well-supported by the study results.

·        There is a good synthesis of the main findings and their implications.

Suggestions for Improvement:

·        Include a brief mention of the study limitations in the conclusions.

·        Discuss more explicitly how the findings compare with previous studies on brucellosis.

7.     Contributions and Limitations

Strengths:

·        The study's contributions are well-articulated.

·        Limitations are acknowledged transparently.

Weaknesses:

·        Contributions and limitations are not concentrated in a specific section at the end of the manuscript.

Suggestions for Improvement:

·        Create a dedicated section at the end of the manuscript that concisely summarizes the contributions and limitations.

·        Discuss in more detail how the identified limitations may have influenced the results.

·        More explicitly state the implications for future research in the conclusions.

8.     References

Strengths:

·        The references cover essential topics related to the study.

·        There is a diversity of sources, including articles on clinical, epidemiological, and methodological aspects.

Weaknesses:

·        Less than 50% of the references are from the last five years.

·        There are few studies specifically on coping with brucellosis.

Suggestions for Improvement:

·        Include more recent studies, especially on brucellosis in China and coping strategies in infectious diseases.

·        Add more references on the Chinese cultural context and its impact on disease coping.

·        Include more studies comparing coping with brucellosis to other chronic infectious diseases.

·        Add more references on information cocoons, given that this is an important concept in the results.

·        Include more recent qualitative studies on patient experiences with chronic diseases.

·        Add recent systematic reviews or meta-analyses on brucellosis or coping with chronic diseases.

III. Final Recommendations

1.     Literature Update:

·        Conduct a comprehensive review of the most recent literature on brucellosis and coping strategies in chronic infectious diseases.

·        Incorporate more studies from the last five years to reflect the current state of knowledge in the field.

2.     Cultural Contextualization:

·        Expand the discussion on the Chinese cultural context and how it influences the process of coping with brucellosis.

·        Include more specific references on health practices and cultural beliefs in China that may affect disease coping.

3.     Methodological Depth:

·        Provide more details on the theoretical saturation process and how it was determined.

·        Explain in more detail how the focus group was conducted and integrated with individual interviews.

·        Describe the process of developing the interview topic guide.

4.     Comparative Analysis:

·        Include a more in-depth discussion on how this study's findings compare with previous specific research on brucellosis.

·        Compare the identified coping strategies with those observed in other chronic infectious diseases.

5.     Implications and Applications:

·        Discuss more explicitly the implications of the findings for clinical practice and health policies.

·        Explore how the results might be applied in different cultural contexts or health systems.

6.     Limitations and Future Research:

·        Create a dedicated section at the end of the manuscript that concisely summarizes the contributions and limitations.

·        Discuss in more detail how the identified limitations may have influenced the results.

·        Delineate specific directions for future research, based on the study's findings and limitations.

7.     Manuscript Structure:

·        Consider reorganizing some sections to improve logical flow, such as including a dedicated section on contributions and limitations at the end of the manuscript.

8.     Conceptual Expansion:

·        Deepen the discussion on the concept of information cocoons, providing more theoretical context and practical examples.

·        Explore in more detail how the concept of "compelled metamorphosing" relates to existing theories on adaptation to chronic diseases.

9.     Ethical Implications:

·        While ethical aspects have been adequately addressed, consider a more in-depth discussion on the specific ethical challenges of conducting research with patients with infectious diseases.

10.  Visual Presentation:

·        Consider including a diagram or visual model that synthesizes the process of "compelled metamorphosing" and its relationships with the identified sociocultural factors.

Author Response

(The authors gave the same response as above.)

Reviewer 3 Report

Comments and Suggestions for Authors

Dear Editor,

Thank you for the opportunity to review this manuscript. This manuscript reports a qualitative study regarding the process of coping among patients with brucellosis in China. The manuscript is well written, the background is clear, the text provides sufficient information for qualitative studies, and the flow is easy to read. To enhance the clarity of the manuscript, I have a couple of comments below: 

  1. Methods: in the data collection part (page 3 lines 103–107), it is mentioned that face-to-face in-dept interviews and one focus group discussion were conducted by MZ, XS, or BF. This statement implies that different participants were interviewed by different researchers, which may lead to different ways of interviewing and probably different outcomes. How did the authors maintain the consistency of the interview protocol and outcomes? Please explain.
  2. Methods: information on page 3 lines 116–118 about the number of participants and length of interviews should belong to the results section. I suggest the authors move that information to the result sections.
  3. Methods: How do the authors maintain the robustness and trustworthiness of this study? Please add an additional section about the trustworthiness of the study.
  4. Results: Before presenting sub-sections 3.1, 3.2, and so on, please provide a narrative explanation of how the results section will be organized, as provided in the abstract. Doing so will help readers understand the flow of the results and what is presented in the result section. 
  5. Results: page 8 line 257, please bold the phrase “withdrawal from social circles” as it is one of the sub-categories.
  6. Results: page 10 line 377, please bold the phrase “Family responsibility.”

Author Response

(The authors gave the same response as above.)
